# The Mixture of Gotu Kola, Cnidium Fruit, and Goji Berry Enhances Memory Functions by Inducing Nerve-Growth-Factor-Mediated Actions Both In Vitro and In Vivo

**DOI:** 10.3390/nu12051372

**Published:** 2020-05-11

**Authors:** Jin Gyu Choi, Zahra Khan, Seong Min Hong, Young Choong Kim, Myung Sook Oh, Sun Yeou Kim

**Affiliations:** 1Department of Oriental Pharmaceutical Science and Kyung Hee East-West Pharmaceutical Research Institute, College of Pharmacy, Kyung Hee University, 26, Kyungheedae-ro, Dongdaemun-gu, Seoul 02447, Korea; choijg2002@khu.ac.kr; 2College of Pharmacy and Gachon Institute of Pharmaceutical Science, Gachon University, 191, Hambakmoe-ro, Yeonsu-gu, Incheon 21936, Korea; zahra.khan37@gmail.com (Z.K.); hongsm0517@gmail.com (S.M.H.); 3College of Pharmacy and Research Institute of Pharmaceutical Science, Seoul National University, Seoul 08826, Korea; youngkim@snu.ac.kr; 4Department of Life and Nanopharmaceutical Sciences, Graduate School, Kyung Hee University, 26, Kyungheedae-ro, Dongdaemun-gu, Seoul 02447, Korea

**Keywords:** Nerve growth factor, memory enhancing, KYJ, Gotu Kola, Cnidium fruit, Goji berry

## Abstract

Nerve growth factor (NGF), a typical neurotrophin, has been characterized by the regulation of neuronal cell differentiation and survival involved in learning and memory functions. NGF has a main role in neurite extension and synapse formation by activating the cyclic adenosine monophosphate-response-element-binding protein (CREB) in the hippocampus. The purpose of this study was to determine whether a mixture of Gotu Kola, Cnidium fruit, and Goji berry (KYJ) enhances memory function by inducing NGF-mediated actions both in vitro and in vivo. The KYJ combination increased NGF concentration and neurite length in C6 glioma and N2a neuronal cells, respectively. Additionally, we discovered memory-enhancing effects of KYJ through increased NGF-mediated synapse maturation, CREB phosphorylation, and cell differentiation in the mouse hippocampus. These findings suggest that this combination may be a potential nootropic cognitive enhancer via the induction of NGF and NGF-dependent activities.

## 1. Introduction

According to recent reports, the population of patients that suffer from dementia has gradually increased worldwide [1]. The loss of memory and cognitive functions is one of the major symptoms of dementia, is a leading cause of death in people over the age of 65, and is caused by various factors including aging, Alzheimer’s disease (AD), and vascular damage [2]. The hippocampus is a critical brain region in memory function, and its shrinkage due to neuronal cell loss is a histological feature of patients with dementia [3]. The cholinergic system, which contains cholinergic neurons and acetylcholine as a neurotransmitter, plays a vital role in the hippocampus and is mediated by neurotrophins such as nerve growth factor (NGF), which is essential for hippocampal plasticity and memory consolidation [4,5].

The influence of nootropics, which are well-known compounds or herbal supplements that enhance mental performance, including memory function, has been widely studied in the search for brain-boosting agents or therapeutic candidates for memory impairment [6]. There are two types of nootropics: synthetic compounds such as piracetam, and natural or herbal nootropics such as ginkgo leaves. Some natural nootropics act as modulators of neurotransmitter release, neuroprotective agents, and as an energy booster in the brain [7,8]. In addition, chronic treatment with nootropics has been associated with significantly enhanced actions on memory function through the increase of NGF expression in the hippocampus [9]. Thus, a natural nootropic that is involved in NGF synthesis might be a potential cognitive enhancer.

As a multi-herbal mixture, KYJ is composed of three plants: Gotu Kola (the whole plant of *Centella asiatica* (L.) Urb.), Cnidium fruit (the fruit of *Cnidium monnieri* (L.) Cuss.), and Goji berry (the fruit of *Lycium barbarum* (L.)). Growing evidence shows that Gotu Kola extract ameliorated memory deficits in 5XFAD AD mice [10], enhanced memory retention [11], protected brain neurons from oxidative stress or amyloid-beta (Aβ) neurotoxicity [12,13], and prevented hippocampal damage in diabetic rats [14]. Additionally, Cnidium fruit extract was shown to attenuate scopolamine-induced amnesia [15]. In the case of Goji berry, chronic administration of its extract enhanced cognitive performance in young healthy people [16] and reduced memory impairment through the reduction of hippocampal Aβ_1–42_ levels in APP/PS1 double-transgenic AD mice [17].

In the present study, we aimed to investigate whether KYJ, the combination of three herbal extracts, could affect NGF synthesis and memory performance as a potential natural nootropic both in vitro and in vivo.

## 2. Materials and Methods 

### 2.1. Materials

Dulbecco’s modified Eagle’s medium (DMEM), supplemented with fetal bovine serum (FBS) and penicillin–streptomycin, was provided by Invitrogen (Carlsbad, CA, USA). Goat polyclonal anti-doublecortin (DCX) and rabbit polyclonal anti-phospho-cAMP-response-element-binding protein (CREB) (pCREB) were purchased from Santa Cruz Biotechnology (Santa Cruz, CA, USA). Biotinylated horse anti-goat antibody, biotinylated goat anti-rabbit antibody, and avidin–biotin complex (ABC) were purchased from Vector Labs (Burlingame, CA, USA). Paraformaldehyde (PFA), 3,3-diaminobenzidine (DAB), sucrose, phosphate-buffered saline (PBS), 30% hydrogen peroxide (H_2_O_2_), lipopolysaccharide (LPS), 3-(4,5-dimethylthiazol-2-yl)-2,5-diphenyl-tetrazolium bromide (MTT), dimethyl sulfoxide (DMSO), piracetam, dibutylphthalate polystyrene xylene (DPX) histomount medium, and mouse anti-synaptophysin were purchased from Sigma-Aldrich (St. Louis, MO, USA). Cerebrolysin was purchased from Ever Neuro Pharma (GMBH, Austria). The enzyme-linked immunosorbent assay (ELISA) development kit for NGF was purchased from R&D Systems (Minneapolis, MN, USA).

### 2.2. Preparation of KYJ

A sample of each plant (2.7 kg) was crushed to a powder and mixed together in a 1:1:1 ratio, extracted with 8 L of 70% ethanol at room temperature for 72 h (three times), and concentrated using a rotary evaporator, resulting in a 405 g ethanol-soluble extract. The yield value of the mixture was 15%.

### 2.3. Cell Culture

In this study, three cell lines—C6 glioma, BV2 murine microglia, and mouse neuro2a (N2a)—were used to study the neuroprotective and anti-neuroinflammatory effects of KYJ. The C6 glioma and BV2 murine microglia were purchased from the Korean Cell Line Bank (Seoul, Korea), and the N2a cells was obtained from American Type Culture Collection (Manassas, VA, USA). All cell lines were maintained in DMEM supplemented with 10% heat-inactivated FBS, 1% penicillin (1 × 10^5^ U/L), and streptomycin (100 mg/L) in a humidified incubator with 5% CO_2_ at 37 °C.

### 2.4. Measurement of NGF Concentration

Measurement of NGF concentration in vitro was conducted according to previously reported methods [18,19]. For the quantitative measurement of secreted NGF, C6 glioma cells were seeded in a 24 well plate at a density of 1 × 10^5^ cells/well and incubated for 24 h. The cells were treated with a specified concentration of the sample for 24 h, and the conditioned medium was collected and centrifuged. The NGF produced in C6 glioma culture supernatants were measured using a competitive ELISA kit in accordance with the manufacturer’s protocol.

### 2.5. Measurement of Neurite Outgrowth

To measure the effect of KYJ on neurite outgrowth, N2a cells were used according to previously reported methods [18,20]. The N2a cells were seeded onto 6 well plates at a density of 1 × 10^4^ cells/well and treated with the KYJ extract for 24 h. These cells ceased to proliferate and began to differentiate, as evidenced by neurite outgrowth, in response to serum starvation, cerebrolysin, or growth factors such as neurotrophins and glial-cell-derived neurotrophic factor family ligands. Neurite lengths of N2a cells were measured using an IncuCyte imaging system (Essen Instruments, Ann Arbor, MI, USA).

### 2.6. Animals and Sample Treatment

Animal treatment and maintenance were carried out in accordance with the Principles of Laboratory Animal Care (NIH publication No. 85–23, revised 1985) and the Animal Care and Use Guidelines of Kyung Hee University, Seoul, Korea (approval number: KHUASP(SE)-19-016). Male ICR (Institute of Cancer Research) mice (5 weeks, 23–25 g) were purchased from the Daehan Biolink Co. Ltd (Eumseong, Korea). Animals were randomly housed, had free access to water and food, and were maintained under a constant temperature (23 ± 1 °C), humidity (60 ± 10%), and a 12 h light/dark cycle. The mice were randomly divided into three groups (*n* = 10/group): (1) a vehicle-treated group, (2) a piracetam (200 mg/kg/day)-treated group, and (3) a KYJ (200 mg/kg/day)-treated group. The KYJ was dissolved in normal saline and administered orally once a day for 16 days. Piracetam was used as a positive control and was administered intraperitoneally for 16 days. An equal volume of normal saline was administered to the vehicle-treated group.

### 2.7. Behavioral Test

The Y-maze test was performed according to previously reported methods [21]. The Y-maze apparatus is composed of three equal angle arms. One hour after the administration, mice were placed in one arm, and were allowed to freely explore the maze. The spontaneous alternation and total entries were manually measured for each mouse over an 8 min period through video analysis. An actual alternation was defined as entry into all three arms consecutively (i.e., ABC, CAB, or BAC but not ABA). The percentage of alternations was calculated as shown by the following equation: ((the number of alternations)/(the total number of arm entries − 2)) × 100.

The step-through passive avoidance test was conducted according to a previously modified method [22]. Briefly, the bright box (21 × 21 × 21 cm) contained a 50 W electric lamp, while the floor of the dark box (21 × 21 × 21 cm) consisted of 2 mm stainless steel rods spaced 1 cm apart. One hour after the administration, mice were placed in the bright compartment, and the door separating the two compartments was opened 10 s later. After the mouse fully entered the dark chamber, the guillotine door was closed and an electrical foot shock (0.4 mA) was delivered through the grid floor for 3 s (acquisition trial). Twenty-four hours after the acquisition trial, each mouse was placed in the bright chamber for a retention trial. Latency was defined as the time it took for a mouse to enter the dark chamber after the door opened (retention trial). Latency time was recorded for up to 600 s.

### 2.8. Brain Tissue Preparation

We performed perfusion to obtain whole mouse brain samples without blood in accordance with previously reported methods [23,24]. Twenty-four hours after the behavior tests, mice were immediately anesthetized and transcardially perfused with 0.05 M PBS, and then fixed with 4% cold PFA in a 0.1 M phosphate buffer (*n* = 6/group). Brains were removed and post-fixed in 0.1 M phosphate buffer containing 4% PFA overnight 4 °C, and then immersed in a solution containing 30% sucrose in 0.05 M PBS for cryoprotection. Serial 30 µm thick coronal sections were cut on a freezing microtome (Leica, Nussloch, Germany) and stored in cryoprotectant (25% ethylene glycol, 25% glycerol, 0.05 M phosphate buffer) at 4 °C until their use for immunohistochemistry.

### 2.9. Immunohistochemistry

Brain sections were serially collected with five to six tissues per group from bregma −1.94 mm to −2.30 mm according to the mouse brain atlas [25]. The free-floating sections were rinsed in PBS buffer and treated with 1% H_2_O_2_ for 15 min. They were incubated with goat anti-DCX (1:100 dilution), rabbit anti-pCREB (1:100 dilution), and mouse anti-synaptophysin (1:1000 dilution) overnight at 4 °C in the presence of 0.3% triton X-100. After rinsing in PBS buffer, the sections were incubated with a biotinylated anti-goat, anti-rabbit, and anti-mouse IgG (1:200 dilution) for 70 min, and then with ABC (1:100 dilution) for 1 h at room temperature. Peroxidase activity was visualized by incubating sections with DAB in 0.05 M tris-buffered saline (pH 7.6) and 0.03% H_2_O_2_. The reaction time of DAB staining was applied equally to each marker to be stained, and the reaction time for each marker was as follows; DCX staining for 70 sec, pCREB staining for 40 sec, and synaptophysin staining for 30 sec. After several rinses with PBS, sections were mounted on gelatin-coated slices, dehydrated, and coverslipped in histomount medium. The number of DCX- and pCREB-positive cells and the neurite length in the dentate gyrus (DG) of the hippocampus were estimated by measuring at 200× magnification using AnalySIS LS Research (Soft Imaging System Ltd., Münster, Germany). The optical density of synaptophysin immunoreactivity in the CA3 region was analyzed with ImageJ software (Bethesda, MD, USA). The images were photographed at 200× magnification using an optical light microscope (Olympus Microscope System BX51; Olympus, Tokyo, Japan) equipped with a 20× objective lens.

### 2.10. Statistical Analysis

All statistical parameters were calculated using GraphPad Prism 5.0 software. Values were expressed as the mean ± S.E.M. Data were analyzed by Student’s t-test. Differences with a *p*-value less than 0.05 were considered statistically significant.

## 3. Results

### 3.1. KYJ Increased NGF Production in C6 Glioma 

We investigated whether KYJ induced NGF production. We found that KYJ (124.36 ± 1.95% at 10 µg/ml; 159.56 ± 7.14% at 25 µg/mL) significantly increased NGF production in C6 glioma cells, which are representative cells of astrocytes (Figure 1A). We further found that KYJ showed no cytotoxicity in C6 glioma cells (Figure 1B). A potent NGF stimulator, 6-shogaol, was used as the positive control [26].

### 3.2. KYJ Induced Neurite Outgrowth in N2a Cells

The increase of NGF production has previously been directly correlated with the enhancement of neurite outgrowth and the differentiation of neuronal cells [27]. With this in mind, N2a cells were treated with KYJ (25 µg/mL) and analyzed for the induction of neurite outgrowth. As shown in Figure 2, KYJ extract induced a significant increase in the length of neurites on the cell bodies of N2a. Cerebrolysin was used as a positive control. KYJ treatment increased neurite length and outgrowth compared with untreated cells and cerebrolysin, respectively.

### 3.3. KYJ Induced Memory Enhancement in Mice

To investigate whether KYJ enhanced memory function, we conducted two behavioral tests after administration. First, spatial memory performance was assessed by the spontaneous alternation (%) in the Y-maze test. The KYJ-treated group (72.88 ± 1.99%; ***p* < 0.01) showed a significant increase in spontaneous alternation compared to the vehicle-treated group (61.07 ± 3.28%). Additionally, the piracetam-treated group showed significant elevation of spontaneous alternation (72.56 ± 3.04%; **p* < 0.05 vs. vehicle-treated group; Figure 3A).

Fear-conditioned memory functions were assessed by analyzing the retention latency time of the mice in the passive avoidance test. There was no difference in the latency time between groups in the acquisition trial, but the retention latency time in the KYJ-treated group (353.62 ± 41.31 sec; ***p* < 0.01) and piracetam-treated group (398.88 ± 58.39 sec; ***p* < 0.01) was significantly longer than that of the vehicle-treated group (178.51 ± 21.46 sec; Figure 3B). These results indicate that the administration of KYJ could enhance memory function.

### 3.4. KYJ Triggered the Phosphorylation of CREB in the Mouse Hippocampus

The activation of CREB via its phosphorylation plays a key role in memory enhancement [28]; thus, we investigated the effects of KYJ on hippocampal CREB activation in mice by analyzing the number of pCREB-immunostained cells in the granule cell layer (GCL) region. As shown in Figure 4, the quantification results for pCREB-positive cells exhibited a marked elevation following treatment with KYJ (12,835.93 ± 1192.34 cells/mm^3^; **p* < 0.05) compared to the vehicle-treated group (9072.62 ± 1027.09 cells/mm^3^). The piracetam-treated group further showed a similar pattern to the KYJ-treated group (12,481.03 ± 1072.52 cells/mm^3^; **p* < 0.05). These findings show that the memory-enhancing effects of KYJ may be closely related to its actions on CREB activation in the hippocampus.

### 3.5. KYJ Promoted Synaptic Formation in the Mouse Hippocampus

Synaptophysin, a potential marker for synaptic vesicles in the integration and connectivity between synaptic terminals, is involved in regulating synapse formation in hippocampal neurons, leading to memory consolidation [29,30]. We explored whether KYJ modulates the synapse formation in the mouse hippocampus by measuring the optical density of synaptophysin-positive CA3 regions. The percentage of synaptophysin-positive intensity in CA3 was significantly higher in both the piracetam- (109.33 ± 2.31%; **p* < 0.05) and KYJ-treated groups (108.53 ± 2.92%; **p* < 0.05) than in the vehicle-treated group (Figure 5). These results suggest that KYJ treatment could induce the strengthening of hippocampal synapse formation that contributes to memory enhancement.

### 3.6. KYJ Stimulated Hippocampal Differentiation of Progenitor Cells and Neurite Extension in Mice

Accumulated evidence shows that NGF acts as a potent inducer of hippocampal cell differentiation and neurite extension [31]. Thus, to investigate whether KYJ affects neuronal cell differentiation and neurite extension in the hippocampal regions, we performed immunohistochemical analyses for DCX, which is present in the cytoplasm of immature neurons located in the GCL of the DG in the hippocampus. In this study, the number of DCX-positive cells in the GCL region of KYJ-treated mice was significantly increased by 21,046.67 ± 1981.81 cells/mm^3^ (**p* < 0.05) compared to those of vehicle-treated mice (16,974.54 ± 615.81 cells/mm^3^; Figure 6D). In addition, piracetam or KYJ treatment significantly increased the total sum of neurite length in DCX-positive immature neurons in the GCL by 295.58 ± 4.99 and 293.32 ± 11.45 μm, compared to the vehicle-treated group (259.95 ± 11.95 μm; Figure 6E), respectively. Collectively, these results suggest that KYJ is involved in the neuronal differentiation of immature neurons and their neurite outgrowth in the mouse hippocampus.

## 4. Discussion

This study was the first to demonstrate the memory-enhancing effects of KYJ, composed of three well-known natural nootropics, by inducing NGF-mediated signaling, including CREB activation, synapse formation, and progenitor cell differentiation in the mouse hippocampus. NGF, a neurotrophin factor continuously produced in the hippocampus, is critical for neuronal survival and development, which requires neurite outgrowth [32]. Our in vitro results showed that KYJ treatment effectively stimulated not only the increase of NGF production, but also the extension of neurite length. We found that oral administration of KYJ reinforced memory functions in mice. These results can be explained by the effects of KYJ on the release of NGF, which can modulate hippocampal plasticity which is directly involved in learning and memory abilities [5]. Moreover, NGF triggers the phosphorylation of CREB to stimulate cAMP-response-element-dependent transcription [33]. Treatment with KYJ resulted in a significantly increased activation of CREB, an important transcription factor involved in memory processing in the hippocampus [34,35].

Synaptogenesis, including synapse formation, stabilization, and maturation, plays a central role in learning and memory [36]. Changes in synaptic strength are closely related to plasticity and neurotransmission in the hippocampus [37]. In this study, we demonstrated a remarkable increase of synaptic density in hippocampal regions of KYJ-treated mice through synaptophysin, one of the most widely used markers for hippocampal plasticity and neurotransmission between synapses [38]. Furthermore, the results showed that KYJ increased the number of DCX-positive immature neurons and neurite outgrowth in the hippocampus. We considered the elevated dendrites associated with KYJ treatment to contribute to synaptic enhancement that is linked to boosting memory function. However, further studies are required to examine whether KYJ may influence hippocampal long-term potentiation, directly showing an enhancement of synaptic plasticity.

Several previous studies on the pharmacological effects of the standardized compounds of each herb in KYJ may partly explain our results. It was reported that asiaticoside derived from Gotu Kola prevented memory decline in senescence-accelerated mice by Aβ deposition [39] and attenuated ischemia/reperfusion-induced memory deficits [40]. Osthole, an isolated compound from Cndium fruit, improved hippocampal synaptic plasticity and cognitive functions in AD rats [41], upregulated neurotrophins to promote hippocampal neurogenesis in APP/PS1 transgenic AD mice [42,43], and inhibited hyperphosphorylation of tau in AD models [44]. Osthole further blocked the reduction of NGF levels in experimental autoimmune encephalomyelitis mice [45]. In addition, the effects of betaine in Goji berry on memory function have been demonstrated by accumulated reports [46,47,48]. The pharmacological activities of these compounds may induce the NGF-mediated memory-enhancing effects of KYJ.

In summary, our data showed that the treatment with KYJ, a mixed herbal formula composed of Gotu Kola, Cnidium fruit, and Goji berry, could induce NGF release and improve memory functions via NGF-mediated pharmacological activities in both in vitro and in vivo systems. There were also several limitations in this study; for example, it is admitted that there are a lack of data on the main active components of KYJ extract, or the exact mechanism in the KYJ that enhances memory function when NGF production is induced. Even so, our results are meaningful enough from the perspective that KYJ could be a potential candidate as a cognitive enhancer for the treatment of dementia. Thus, we expect that our findings could contribute to the development of nootropics as one of mechanisms of anti-dementia drugs.

## Figures and Tables

**Figure 1 nutrients-12-01372-f001:**
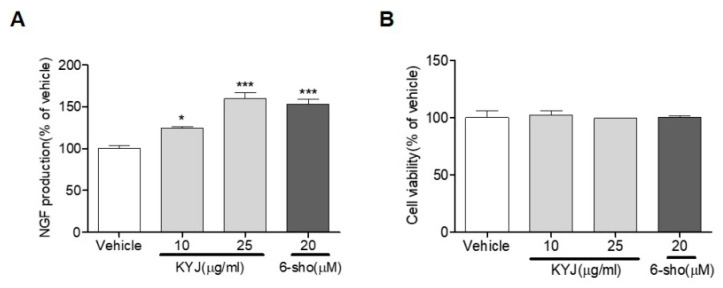
The mixture of Gotu Kola, Cnidium fruit, and Goji berry (KYJ) stimulated the release of nerve growth factor (NGF) in C6 cells. (**A**) Cells were treated with KYJ (10 and 25 µg/mL) or 6-shogaol (20 µM, positive control), and the NGF concentration was measured in culture media. (**B**) The cell viability was expressed as the result of MTT assay. Three independent experiments were performed. **p* < 0.05 and ****p* < 0.001 vs. vehicle-treated group.

**Figure 2 nutrients-12-01372-f002:**
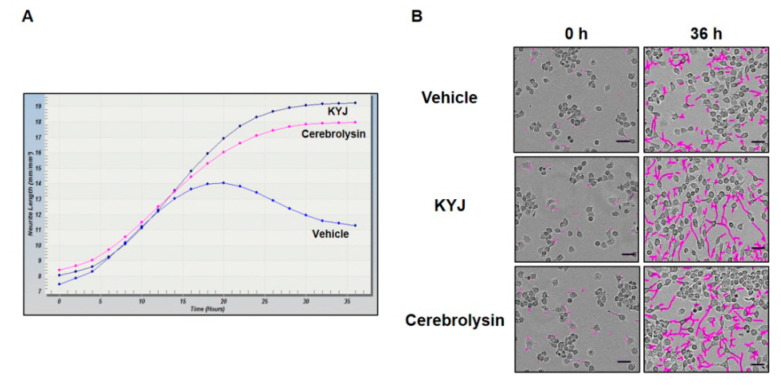
KYJ extended the neurite length of N2a cells. (**A**) The measurement of neurite length was conducted at various time points until 36 h. (**B**) Neurite length was measured and representative images were taken at 0 h and after 36 h. Scale bar = 50 µm.

**Figure 3 nutrients-12-01372-f003:**
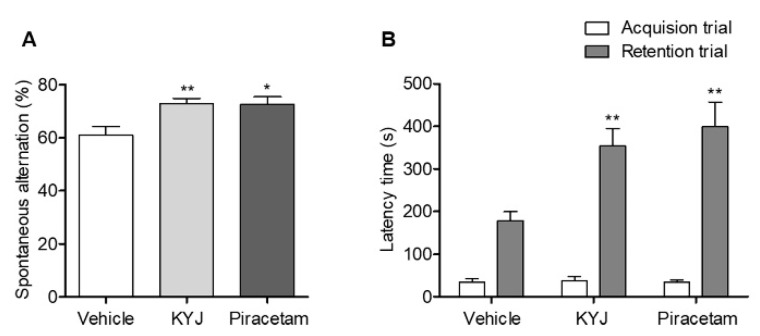
KYJ enhanced memory functions in mice. Y-maze test and passive avoidance test were performed 13 to 15 days after the administration of piracetam or KYJ, respectively. (**A**) The spontaneous alternations (%) in the Y-maze test and (**B**) the latency time (s) to enter the dark compartment during the acquisition trial (white) and 24 h later during the retention trial (gray) in passive avoidance test. **p* < 0.05 and ***p* < 0.01 vs. the vehicle-treated group.

**Figure 4 nutrients-12-01372-f004:**
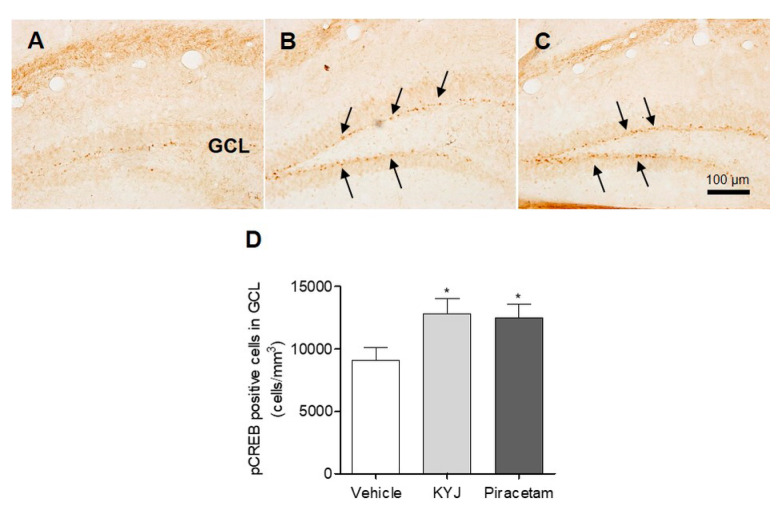
KYJ stimulated the phosphorylation of hippocampal CREB in mice. (**A**–**C**) Representative images of each group are shown: (**A**) vehicle-treated group, (**B**) KYJ 200 mg/kg/day (per os; p.o.) group, (**C**) piracetam 200 mg/kg/day (Intraperitoneal injection; i.p.) group. (**D**) The number of pCREB-positive cells (black arrow) in the GCL region was quantified via stereological counting. **p* < 0.05 vs. the vehicle-treated group.

**Figure 5 nutrients-12-01372-f005:**
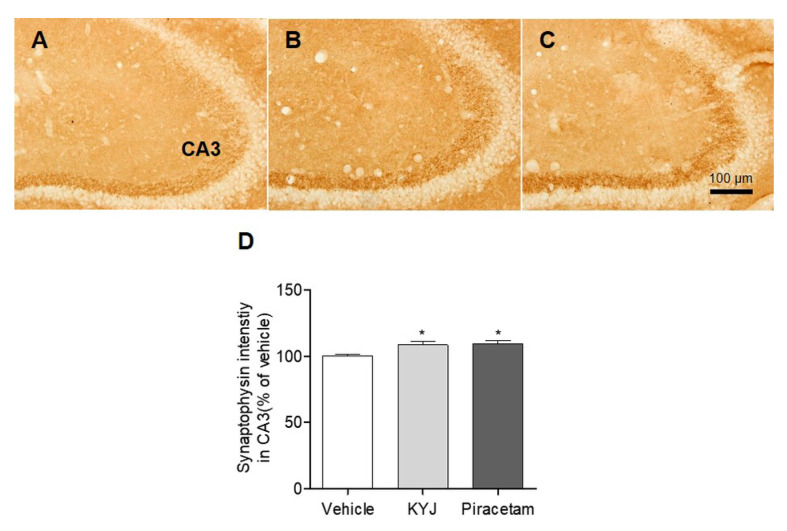
KYJ reinforced synapse formation in the mouse hippocampus. (**A**–**C**) Representative images of each group are shown: (**A**) vehicle-treated group, (**B**) KYJ 200 mg/kg/day p.o. group, (**C**) piracetam 200 mg/kg/day i.p. group. (**D**) The optical density of synaptophysin immunoreactivity in CA3 region was measured. **p* < 0.05 vs. the vehicle-treated group.

**Figure 6 nutrients-12-01372-f006:**
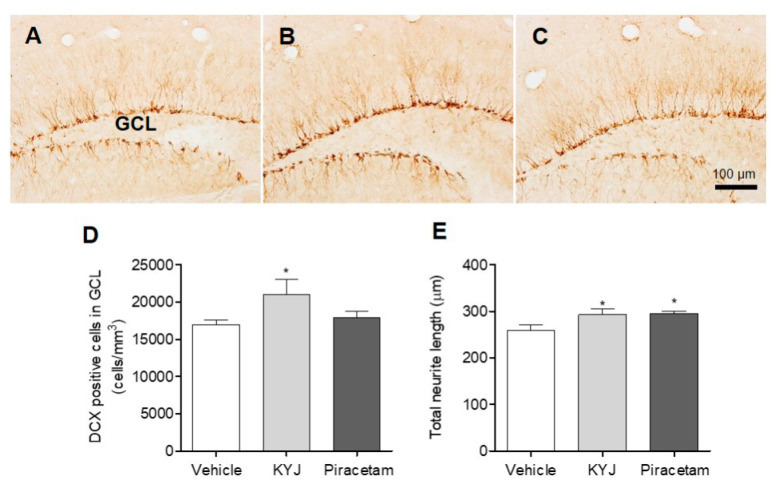
KYJ promoted hippocampal cell differentiation and neurite extension in mice. (**A**–**C**) Representative images are shown: (**A**) vehicle-treated group, (**B**) KYJ 200 mg/kg/day p.o. group, (**C**) piracetam 200 mg/kg/day i.p. group. (**D**) The number of DCX-positive cells in the GCL region was quantified via stereological counting. (**E**) The total neurite length was the sum of DCX-positive cell’s neurite length. **p* < 0.05 vs. the vehicle-treated group.

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
