# Peer review of "The Mixture of Gotu Kola, Cnidium Fruit, and Goji Berry Enhances Memory Functions by Inducing Nerve-Growth-Factor-Mediated Actions Both In Vitro and In Vivo"

_nutrients, 2020, doi:10.3390/nu12051372_

Round 1

Reviewer 1 Report

The present manuscript addresses the impact of a mixture of Gotu Kola, 24 Cnidium fruit, and Goji berry (KYJ) on memory functions, by inducing NGF-mediated actions 25 in vitro and in vivo. This approach is introduced as one possible improvement for dementia therapy. The manuscript is well organized, and the chosen methodological approach is sound. There are a couple of matters authors should consider in a revised version of the manuscript. After changing the manuscript following these major points and providing details in a response letter, this manuscript can be a mature piece for publication in this journal:

  • The research community could better benefit from your study if you provided the limitations of your study and method. In other words, please make clear which aspects could not be analyzed through your method, but which would be of importance for future research in this field.

  • In your material and methods section, you are transparent about your methodological approach including study parameter. I wonder a bit why you do not refer to other similar studies when introducing thresholds and parameters? This would help your readership to better compare the studies working on similar topics.

  • You end up with a quite short discussion section. A summary is missing that brings together the main findings of your research. Could you also discuss in how far your study would be a step towards the improvement of dementia therapy?

Reviewer 2 Report

This manuscript reports the combination use of three natural products to stimulate memory functions in mice, potentially via inducing release of nerve growth factor. Overall, the manuscript is well written and presented and is easy to read and follow. There are, however, some suggestions for improvement to make the results more convincing.

For the in vitro astrocyte model they use the C6 glioma cell line. This is a glioma cell derived from rats. While is known that C6 cells can be stimulated to express NGF, glioma cells are known to have altered characteristics and thus the response to the triple KYJ herbal mixture may not reflect normal astrocyte responses. In addition, it is a rat cell line, while the other experiments use mouse cells and in vivo mouse models. It would be better to use mouse primary astrocytes to confirm the results. Purifying mouse astrocytes is relatively straightforward and can be achieved within a few weeks.

The use of DAB to visualise the antibody stainings for the in vivo experiments is not convincing. If the reaction time is not kept constant, it is very easy for the intensity of the staining to be different, and no regulation of timing of reaction was described in the methods. For example, in Fig 4, the control panel A background appears much lighter than the other panels. Having a stronger reaction would result in more cells being visualised, affecting the cell count in the graph in panel D. Similarly, in Fig 6, the background colour in panel A is fainter than in the other panels, indicating that the reaction time was shorter. Having a stronger staining would result in being able to visualise the fine processes which would alter the quantification of the neurite length shown in panel E. Fluorescent immunostaining is a more convincing quantification method and should be provided to confirm the DAB staining results.

Considering the results together, there is no clear evidence that the triple herbal extract alters NGF expression in vivo and thus the title should be altered as it cannot be concluded that memory function is enhanced by NGF release in vivo.

Minor change:

The first sentence in section 3.2, Line 171, should be altered as some words seem to be missing. Perhaps it should read “The increase of NGF production has previously been directly correlated…”

Round 2

Reviewer 1 Report

The authors addressed the major points of critcism (in Round 1) in an adequate response letter. The line of argumentation is clearer and the manuscript was thoroughly revised. Therefore, I would like to recommend the present version of the manuscript for publication in this journal.

Reviewer 2 Report

To authors have satisfactorily addressed the queries. The new images for Fig 4 and Fig 6 are much improved. The new title is much better.